# Sustainability of an Educational Program on Oral Care/Hygiene Provision by Healthcare Providers to Older Residents in Long-Term Care Institutions: A Follow-Up Study

**DOI:** 10.3390/geriatrics9030084

**Published:** 2024-06-20

**Authors:** Florence M. F. Wong, Wai Keung Leung

**Affiliations:** 1Tung Wah College, Hong Kong SAR, China; 2Faculty of Dentistry, The University of Hong Kong, Hong Kong SAR, China; ewkleung@hku.hk

**Keywords:** caregivers, dental care for aged, education, professional, retraining, health knowledge, attitudes, practice, health services for the aged, long-term care

## Abstract

Background: The importance of oral health in older adults, especially those in long-term care institutions (LTCIs), has been widely recognized. This study aimed to evaluate the sustainability of an oral health educational program (OHEP) for healthcare providers by measuring changes in their knowledge, attitudes, and practice (KAP) towards oral care provision 3 and 6 months after completing the OHEP. Methods: A pragmatic direct care nursing education trial with a control group was conducted to evaluate the sustainability of an OHEP by examining changes in KAP 3 and 6 months after the OHEP. The OHEP comprised both knowledge and skills related to oral care, whereas the control group received standard support in accordance with usual oral care practice. Results: The study included 20 healthcare providers in the intervention group and 20 in the control group. At 6 months post-OHEP, a significant difference in knowledge was observed between the two groups, with the intervention group maintaining a positive effect (mean 13.90). Conversely, the control group showed a significant decline in knowledge (from mean 14.25 to 12.10). Both groups showed an improvement in attitudes regarding oral health, with the intervention group exhibiting better results 3 months post-OHEP. Intervention group participants rated oral care as a higher priority. Conclusions: An OHEP program for LTCI direct care staff provides enhanced knowledge and attitudes toward oral health care. Regular training in direct care and additional support may be needed to sustain optimal effects on oral care practice.

## 1. Introduction

Poor oral health among older people is a global concern [1,2,3], with common problems observed in this population including sore mouth and tongue, gum diseases, tooth loss, and decay [3,4]. In the United States, 17% of the population aged 65 and above have no teeth, and over 95% have decayed, filled, or missing teeth [4]. Oral health management and the prevention of severe conditions such as dental caries, periodontal diseases, oral cancers, and infections are gaining attention [5,6,7,8,9]. Poor oral hygiene contributes significantly to poor oral health, which in turn leads to eating problems, weight loss, speech difficulties, and nutritional problems [3,10,11,12]. Residents in long-term care institutions (LTCIs) with limited self-care abilities, particularly those with cognitive concerns, are more likely to have poor oral hygiene [13,14,15]. Healthcare providers play an important role in maintaining oral health in this population, but barriers such as limited resources, time constraints, and lack of knowledge make oral care a low priority [16]. A systematic review identified poor oral health to be associated with factors such as increasing age, cognitive disorders, edentulousness, dental caries, and periodontal diseases, resulting in poorer oral health-related quality of life [16]. However, poor oral care of older residents has been attributed to inadequate knowledge, unfavorable attitudes, and poor practices in oral care among healthcare providers [17,18,19,20,21].

Several studies have consistently found that healthcare providers in LTCIs have limited oral care knowledge and neglect oral care practices due to tight schedules and poor perceptions [18,19,20]. To address this, educational programs on oral health and oral care have been recommended and investigated [22,23,24,25,26,27]. Numerous studies have demonstrated the effectiveness of these programs in improving healthcare providers’ knowledge, attitudes, and practice (KAP) [22,23,24,25,26,27]. However, variations in program designs pose challenges in assessing their overall effectiveness, and many studies have only evaluated short-term effects without considering program sustainability. For instance, Konstantopoulou et al. [24] conducted a quasi-experimental study to assess the impact of an oral care educational program on formal caregivers in a nursing home. The intervention group and control group consisted of 28 caregivers and 27 caregivers, respectively. The program included theoretical (PowerPoint presentations on oral health) and practical (demonstrations of oral and denture hygiene practices) components. The results indicated that the program effectively improved the knowledge and attitudes of caregivers toward oral health in nursing home residents, with sustained effects for two months. Similarly, Le et al. [26] conducted an interventional study targeting support staff in nursing homes. The program included knowledge about oral health conditions, oral health promotion, daily oral care provision, and oral care decision-making strategies. The study demonstrated that the program improved the staff’s oral care knowledge and led to improvement in the oral conditions of LTCI residents within six months post-implementation. Evaluating the longer-term effects of educational programs on oral health and care for older residents, such as those at three to six months post-implementation, is crucial for ensuring that improvements in KAP are maintained.

To enhance the knowledge and skills of healthcare providers, an oral health educational program (OHEP) was developed to enhance the knowledge and skills of healthcare providers and in turn improve their oral care practices for older residents in LTCIs in the original study [28]. The program utilized materials from the Oral Health Education Division of the Department of Health [29]. The current study is a follow-up study conducted to explore the short-term impact (3 and 6 months post-OHEP) of the program, evaluating its effectiveness in improving healthcare providers’ KAP regarding oral care for older residents. The objectives were to understand the daily care duty priorities of LTCI healthcare providers and to evaluate the short-term effects (3 and 6 months) of the OHEP on the changes in healthcare providers’ KAP of oral care for their older residents. It was hypothesized that (a) the OHEP would be effective in enhancing the KAP of oral care among healthcare providers in LTCIs, and (b) the OHEP would have sustained effects on healthcare providers’ KAP of oral care 3 and 6 months post-OHEP.

## 2. Materials and Methods

### 2.1. Study Design and Participants

The present study was a quasi-experimental longitudinal study with a control design. The research adhered to the STROBE guidelines (Appendix B) [30] to ensure proper reporting and transparency. As described in the original study [28], this study was conducted in two LTCIs, with one designated as the intervention group and the other designated as the control group. The intervention LTCI received the OHEP, whereas the control LTCI received standard support. A minimum of 40 healthcare providers (20 in each group) was required to reach the desired power of 0.85 and a Type I error of 0.05 with an effect size of 0.5, as calculated using the G*Power version 3.1, Universität Kiel, Kiel, Germany [31]. The study was conducted in accordance with the Declaration of Helsinki and approved by the Institutional Ethics Committee of Tung Wah College (reference number REC2021101; date of approval: 23 September 2021).

### 2.2. Oral Health Educational Program (OHEP)

The OHEP utilized in this study comprised two parts conducted over a total of four lessons. These lessons consisted of two educational sessions and two sessions dedicated to demonstrating oral care skills, following established protocols [22,23,24,25,26,27]. The educational component provided fundamental knowledge regarding oral health, specifically addressing common oral issues in older individuals. The skill demonstration portion involved the trainer (PI), an experienced nurse educator, first showcasing the required skills before the healthcare providers practiced these skills through a return demonstration. Each lesson had a duration of approximately one hour and was delivered weekly in a conference room at the LTCI where the study was conducted. The details of the OHEP were described in the original study [28]. Healthcare providers in the control group received the standard support, with oral care performed by direct care staff providers in accordance with usual oral care practices.

### 2.3. Instruments

#### 2.3.1. Survey for Duty Priorities

A survey was designed and administered before the initiation of the OHEP to gain insights into the LTCI care providers’ self-reported duty priorities for older residents in their daily practice. The main duties in their practice included in the survey were the administration of oral medication (AOM), feeding, oral care, showering, and monitoring of vital signs.

#### 2.3.2. KAP of Healthcare Providers in Oral Health and Care for Older Residents

The KAP survey developed by Wong [32] was used in this study. The response options of ‘Yes,’ ‘Don’t know,’ and ‘No’ were employed for knowledge, and the 5-Likert scales ranged from 1 (strongly disagree) to 5 (strongly agree) for attitudes and practice. The KAP scale ranges were 0 to 19, 13 to 65, and 19 to 95, respectively, with a total score range of 32 to 179. Additional information on the survey can be found in the original study [28]. 

### 2.4. Study Procedure

After ethics approval (reference number REC2021101, Tung Wah College) was obtained, two LTCIs were carefully chosen for this study, considering their sizes and the services they provided to older residents and ensuring of the inclusion of the required number of healthcare providers. The two LTCIs were operated by the same organization. The nurse in charge of the main LTCI played a pivotal role in assigning one LTCI as the intervention group and the other as the control group. To ensure smooth coordination, the principal investigator (PI) engaged in discussions with the nurse in charge regarding the schedules for the four-stage data collection (pre- and post-OHEP, as well as 3 months and 6 months post-OHEP) and the implementation of the OHEP. The flow of the study is illustrated in Appendix A.

All healthcare providers were required to complete the demographic form, the survey for duty priorities in five major nursing care areas, and the KAP survey on oral health and oral care for older residents at the pre-OHEP. Following the OHEP, the healthcare providers were requested to complete the KAP survey again. This study focused on the data collection at 3 months and 6 months post-OHEP. All data for this study were collected through face-to-face interviews, with the questionnaires requiring approximately 20 min to complete.

### 2.5. Statistical Analysis

Data analysis was conducted using SPSS v26.0 (IBM Corporation, Armonk, NY, USA). The normality of continuous variables was assessed using skewness statistics and normal Q-Q plots, and no violations of the normality assumption were found. Descriptive statistics were used to summarize the duty priorities and the outcome variables (KAP) at 3 months and 6 months post-OHEP. The associations of the outcome variables (KAP) between the two groups were assessed using Chi-square tests or univariate analysis, including Pearson’s correlation coefficient, an independent samples *t*-test, and one-way analysis of variance (ANOVA), depending on the outcome variables’ measurement level. The Chi-square tests or Fisher’s exact tests were used to evaluate the association between ratings of the duty priorities between the two groups. ANOVA tests were used to assess the associations between the ratings of self-reported duty priorities pre-OHEP and KAP at the four study periods. A repeated-measures ANOVA was conducted to analyze changes in KAP at various study points. Mauchly’s test was performed to assess the assumption of sphericity and determine whether significant changes occurred across the four study periods. Additionally, the test of within-subject effects was used to examine the main effects and interactions among the within-subject factors. All statistical tests were two-sided, and a *p*-value <0.05 was considered statistically significant.

## 3. Results

### 3.1. Participants’ Demographic Characteristics and Duty Priorities of Daily Care

The original study reported the demographic characteristics of healthcare providers in detail [28]. The majority of the 40 healthcare providers participating in the study, with 20 in the intervention group and 20 in the control group, had not previously received oral care training.

Regarding the duty priorities of daily care, oral care was rated as the highest priority in the intervention group, whereas showering was rated as the highest priority, and oral care as the third-highest priority, in the control group.

Moreover, Chi-square tests revealed significant differences (*p* < 0.001) in all duty priorities except for AOM between the two groups. Table 1 presents the healthcare providers’ duty priorities of the five daily care duties.

### 3.2. Differences in KAP between the Two Groups and at Various Study Periods

The results showed that the knowledge of healthcare providers in the intervention group had declined at 3 months post-OHEP (mean scores reduced from 14.00 to 13.30), but no further decline was observed 6 months post-OHEP. On the contrary, knowledge among control group participants declined at both 3 months and 6 months post-OHEP (mean scores reduced from 14.25 to 12.1, *p* < 0.001). Attitudes toward oral health care improved in both groups throughout the study, though no significant differences were observed between the two groups. Oral health care practices improved in both groups (mean scores of intervention group from 42.85 to 50.4 and control group from 48.4 to 50.4) at 6 months post-OHEP. However, according to Mauchly’s test of sphericity analysis, only knowledge in both groups and practice in the control group showed significant trends. The tests of within-subject effects found no significant impact of any study periods on knowledge, attitudes, practice, or overall KAP scores in both groups, implying that the mean scores for all outcome variables were similar across the four study periods in both groups. 

Table 2 presents the knowledge, attitudes, and practice scores between the intervention and control groups at four study periods and the changes in KAP between different study periods.

### 3.3. Associations between Oral Care Duty Priority Ratings at Baseline and KAP at Various Study Periods

The ANOVA tests were conducted to identify associations between oral care duty priority, as rated by the participating healthcare providers at baseline, and their KAP of oral health and care at the four study periods. The self-reported baseline oral care priority among healthcare providers in the control group had a weakly significant association with their practice scores in the post-OHEP period (no intervention was delivered to control participants) [F(1, 18) = 4.56, *p* = 0.047], and the self-reported baseline oral care priority among healthcare providers in the intervention group had a moderately significant association with their attitude scores in the 3-month post-OHEP period [F(1, 18) = 5.46, *p* = 0.031]. However, no significant associations were found between self-reported baseline oral care priorities and other aspects of KAP of oral health and care in either group.

### 3.4. Effects of the OHEP in KAP Changes between the Two Groups

The healthcare providers in the intervention group showed a slight improvement in knowledge, whereas those in the control group demonstrated a further decline in knowledge from the 3-month to 6-month post-OHEP period. The mean change in knowledge in the intervention group was 0.6, and that in the control group was −2.15 (*p* = 0.024). The intervention group had a greater change than the control group in attitudes toward oral health at 3 months post-OHEP, though the difference between the two groups was not statistically significant. Table 3 shows the KAP of healthcare providers in the intervention and control groups at the four study periods.

Though the main effect of the intervention on knowledge scores was significant over the four study periods [F(3, 114) = 6.83, *p* < 0.001], the impact of the intervention on knowledge scores varied over time. The main effect of the intervention on attitude was also significant [F(3, 114) = 8.77, *p* < 0.001) but did not vary over time. The intervention did not reach significant main effects on practice scores.

## 4. Discussion

The primary objectives of this study were to assess the effectiveness and sustainability of the OHEP in improving the KAP of oral health and care for older residents among LTCI healthcare providers at 3 months and 6 months post-OHEP. Only 25% of healthcare providers in the intervention group had prior experience in taking care of older residents in LTCIs, whereas 75% of healthcare providers in the control group had previous experience. Detailed demographic results are provided in the original study [28].

Control group participants were more concerned with assisting in personal care for older residents, whereas those in the intervention group were more concerned with health-related oral care. The assignment of duties for staff is usually performed by nurses who may have preferentially assigned control group participants to personal care duties. It is also possible that healthcare providers in LTCIs have limited awareness or authority to improve the oral conditions of older residents and promote good oral health [33,34,35].

Nevertheless, healthcare providers in the intervention group showed a modest improvement in knowledge after the educational program, which declined at a 3-month follow-up and then was maintained at a 6-month follow-up Meanwhile, the healthcare providers in the control group demonstrated a progressive decline in oral care knowledge. This may be attributed to their personal perceptions and inadequate confidence or experience in providing oral care [28]. In the control group, healthcare providers rated oral care as their third-highest priority. In contrast, the intervention group rated oral care as their highest priority. Oral health educational programs should emphasize the importance of oral health and care for residents of LTCIs [28,36,37,38]. Though educational programs are known to enhance the oral care knowledge of healthcare providers [22,23], this knowledge may not be sustained. Enhancing healthcare providers’ competence and oral care practices may thus require regular training and support, including the provision of structured guidelines, implementation of continuous oral health assessment for residents [39,40,41], and interdisciplinary collaboration between nursing and dentistry.

### Strengths and Limitations

One of the strengths of the study is its longitudinal observational design assessing the effectiveness and sustainability of an OHEP over a 6-month period. However, the study was performed at two institutions and had a small sample size, thereby limiting the generalizability of the findings.

## 5. Conclusions

Maintaining the oral health of older individuals is a critical aspect of gerontological care. Our study emphasizes the importance of healthcare providers’ KAP regarding oral health and care for older residents of LTCIs. The implementation of OHEPs is crucial for providing healthcare professionals with the necessary training on oral health and care for older residents in LTCIs. This study revealed that an OHEP for LTCI healthcare providers facilitates enhanced knowledge and attitudes of oral health and care. Regular training and further education may be needed to achieve similar improvements in and maintain a longer effect on oral care practices. 

## Figures and Tables

**Table 1 geriatrics-09-00084-t001:** Participants’ self-reported pre-intervention five major daily care duties.

		Self-Reported Priority ^1^ (%)		Chi-Square Statistic ^2^	
Task		Intervention (*n* = 20)	Control (*n* = 20)	χ^2^	*p*
AOM					
	1	3 (15)	0 (0)		
	2	2 (10)	1 (5)		
	3	2 (10)	0 (0)		
	4	11 (55)	3 (15)		
	5	2 (10)	16 (80)	3.14	0.091
Feeding					
	1	0 (0)	2 (10)		
	2	4 (20)	14 (70)		
	3	10 (50)	3 (15)		
	4	6 (30)	1 (5)		
	5	0 (0)	0 (0)	14.40	<0.001
Oral care					
	1	16 (80)	1 (5)		
	2	0 (0)	3 (15)		
	3	4 (20)	16 (80)		
	4	0 (0)	0 (0)		
	5	0 (0)	0 (0)	14.40	<0.001
Showering					
	1	1 (5)	16 (80)		
	2	0 (0)	2 (10)		
	3	1 (5)	1 (5)		
	4	1 (5)	0 (0)		
	5	17 (85)	1 (5)	28.97	<0.001
Monitoring of vital signs					
	1	0 (0)	1 (5)		
	2	14 (70)	0 (0)		
	3	3 (15)	0 (0)		
	4	2 (10)	16 (80)		
	5	1 (5)	3 (15)	18.03	<0.001

AOM: Administration of oral medication; ^1^ Five major daily care duties—1: highest priority, 5: lowest priority; ^2^ First and second priorities vs. 3rd, 4th, and 5th priorities.

**Table 2 geriatrics-09-00084-t002:** KAP score differences between the two groups and at various study time points.

KAP	Control (*n* = 20)		Intervention (*n* = 20)		Independent *t*-test			Comparison ^1^	Mauchly’s Test of Sphericity		Tests of Within-Subject Effects ^2^	
	Mean	SD	Mean	SD	t	*p*	95% CI		Control	Intervention	Control	Intervention
K												
1 ^3^	17.25	2.15	13.55	4.51	3.31	0.030	1.41 to 5.99	0 vs. 1			F(1, 18) = 0.20, *p* = 0.657	F(1, 18) = 0.20, *p* = 0.657
2 ^3^	14.30	2.13	14.00	1.86	0.47	0.638	−0.98 to 1.58	1 vs. 2			F(1, 18) = 0.43, *p* = 0.520	F(1, 18) = 0.89, *p* = 0.358
3	14.25	1.97	13.30	1.81	1.59	0.121	−0.26 to 2.16	2 vs. 3			F(1, 18) = 0.91, *p* = 0.354	F(1, 18) = 0.04, *p* = 0.841
4	12.10	0.55	13.90	1.92	−4.04	<0.001	−2.70 to −0.90	0, 1, 2, 3	ꭓ^2^ (5) = 55.67, *p* < 0.001	ꭓ^2^ (5) = 17.06, *p* = 0.004	F(3, 54) = 0.35 *p* = 0.603	F(3, 54) = 0.38, *p* = 0.662
A												
1 ^3^	49.30	4.84	47.10	4.78	1.45	0.156	0.88 to5.28	0 vs. 1			F(1, 18) = 0.89, *p* = 0.361	F(1, 18) = 0.65, *p* = 0.433
2 ^3^	51.35	4.32	51.00	4.21	0.26	0.797	−2.38 to 3.08	1 vs. 2			F(1, 18) = 0.05, *p* = 0.827	F(1, 18) = 2.95, *p* = 0.104
3	52.30	4.86	53.15	5.74	−0.51	0.616	−4.26 to 2.56	2 vs. 3			F(1, 18) = 1.12, *p* = 0.306	F(1, 18) = 0.17, *p* = 0.689
4	50.65	2.64	51.60	4.84	−0.77	0.447	−3.47 to 1.57	0, 1, 2, 3	ꭓ^2^ (5) = 6.57, *p* = 0.256	ꭓ^2^ (5) = 10.62, *p* = 0.060	F(3, 54) = 0.84, *p* = 0.454	F(3, 54) = 0.82, *p* = 0.457
P												
1 ^3^	48.40	5.62	42.85	5.35	3.20	0.003	2.04 to 9.06	0 vs. 1			F(1, 18) = 3.08, *p* = 0.100	F(1, 18) = 1.76, *p* = 0.202
2 ^3^	51.05	5.76	46.45	7.44	2.19	0.035	0.33 to 8.87	1 vs. 2			F(1, 18) = 0.40, *p* = 0.534	F(1, 18) = 0.28, *p* = 0.606
3	50.00	6.34	45.40	7.10	2.16	0.037	0.29 to 8.91	2 vs. 3			F(1, 18) = 0.23, *p* = 0.640	F(1, 18) = 0.67, *p* = 0.424
4	50.40	5.32	45.00	6.38	2.91	0.006	1.64 to 9.16	0, 1, 2, 3	ꭓ^2^ (5) = 13.36, *p* = 0.021	ꭓ^2^ (5) = 7.26, *p* = 0.203	F(3, 54) = 1.49, *p* = 0.241	F(3, 54) = 0.69, *p* = 0.529
KAP												
1 ^3^	114.95	8.50	103.50	12.54	3.38	0.002	4.56 to 18.34	0 vs. 1			F(1, 18) = 3.14, *p* = 0.096	F(1, 18) = 0.51, *p* = 0.484
2 ^3^	116.70	7.55	111.45	10.68	1.80	0.082	−0.67 to 11.19	1 vs. 2			F(1, 18) = 0.04, *p* = 0.844	F(1,18) = 0.72, *p* = 0.409
3	116.55	8.91	111.85	12.43	1.37	0.178	−2.22 to 11.62	2 vs. 3			F(1, 18) = 0.00, *p* = 0.958	F(1,18) = 0.02, *p* = 0.888
4	113.15	6.67	110.50	11.31	0.90	0.374	−3.34 to 8.64	0, 1, 2, 3	ꭓ^2^ (5) = 7.38, *p* = 0.195	ꭓ^2^ (5) = 8.11, *p* = 0.151	F(3, 54) = 1.40, *p* = 0.254	F(3,54) = 0.33, *p* = 0.743

^1^ 0 = Pre-OHEP, 1 = post-OHEP, 2 = 3 months post-OHEP, 3 = 6 months post-OHEP; ^2^ Adjusted for the experience of working at long-term care institutions; ^3^ Data were presented in an earlier publication from this group [28]; ‘post-intervention’ for the control group = no intervention.

**Table 3 geriatrics-09-00084-t003:** Knowledge, attitudes, and practice of healthcare providers in intervention and control groups at the 4 study periods.

	Control (*n* = 20; Mean ± SE)	Intervention (*n* = 20; Mean ± SE)	
KAP Scores			*p*
Knowledge (K)			
Pre- to post-intervention ^1^	−2.95 ± 0.81	0.45 ± 1.20	0.024
Post- to 3 months post-intervention	−0.05 ± 0.11	−0.70 ± 0.62	0.306
3 to 6 months post-intervention	−2.15 ± 0.45	0.60 ± 0.48	<0.001
Attitudes (A)			
Pre- to post-intervention ^1^	2.05 ± 1.46	3.90 ± 1.45	0.374
Post- to 3 months post-intervention	0.95 ± 0.93	2.15 ± 1.43	0.487
3 to 6 months post-intervention	−1.65 ± 1.32	−1.55 ± 1.52	0.961
Practice (P)			
Pre- to post-intervention ^1^	2.65 ± 1.62	3.60 ± 2.01	0.715
Post- to 3 months post-intervention	−1.05 ± 0.90	−1.05 ± 2.31	1.000
3 to 6 months post-intervention	0.40 ± 2.04	−0.40 ± 1.41	0.749
Overall KAP			
Pre- to post-intervention ^1^	1.75 ± 2.15	7.95 ± 3.72	0.157
Post- to 3 months post-intervention	−0.15 ± 1.40	0.40 ± 3.54	0.886
3 to 6 months post-intervention	−3.40 ± 2.57	−1.35 ± 2.99	0.606

SE: standard error of the mean; ^1^ Data were presented in an earlier publication from this group [28]; ‘post-intervention’ for the control group = no intervention.

## Data Availability

The data presented in this study are available on request from the corresponding author. The data are not publicly available due to confidentiality reasons regarding the participants.

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
