# Peer review of "Sustainability of an Educational Program on Oral Care/Hygiene Provision by Healthcare Providers to Older Residents in Long-Term Care Institutions: A Follow-Up Study"

_geriatrics, 2024, doi:10.3390/geriatrics9030084_

Round 1

Reviewer 1 Report

Comments and Suggestions for Authors

Dear Authors,

The present research article, entitled “Sustainability of an educational programme on knowledge, attitudes, and practice of oral care/hygiene provision by healthcare providers to older residents in long-term care institutions: A follow up study”, aims to 1) to evaluate the effectiveness of the OHEP on the KAP of oral care among LTCI healthcare providers for their older residents; and b) to assess the short-term (6 months) sustainability effects of the OHEP on the changes in KAP of oral care among LTCI healthcare providers for their older residents.

The main strength of this study is that it aims to examine the sustainability of the effects of the education program on the KAP of healthcare providers and to contribute to the development of effective strategies for promoting the oral health of older residents in LTCIs.

In general, I believe that the topic and approach of this article is timely and of interest to the readers of Geriatrics. However, I believe that some issues should be included to improve the quality of the manuscript.

Abstract

·         Include what the intervention consisted of and what the control group did.

·         It is recommended to include the implications of the differences found between the groups.

Introduction

·         How long is the long term of your study? to be understood in terms of previous literature and the objectives you propose at the end of the introduction, it is recommended to anticipate the information in terms of the "long term".

·         Please add the study hypothesis, which was missing in the manuscript.

Methods

·         It is recommended (despite alluding to the original study) that information on the intervention and standard support be included to understand the study. Also, information on the instruments.

·         What was the procedure? How was the questionnaire applied: online, face-to-face...? How long did it take to complete? How long did the intervention last and how was it carried out?

·         Were there exclusion criteria?

·         On what dates did the data collection and intervention take place?

Results

·         It is recommended to include an explanatory note in Table 1, with the meaning from 1 to 5.

Discussion

·         It is recommended to start the discussion with the objectives of the study.

·         An example of the most relevant results of the study could be put in the conclusions.

 Best regards.

Author Response

Dear Reviewer,

Thank you very much for your valuable comments. We have reviewed and revised the manuscript according to your comments. Attached please find the replies to your comments. You may find the revisions are highlighted in yellow in the revised manuscript. 

Thank you again for your kind review.

Regards,

Florence

Reviewer 2 Report

Comments and Suggestions for Authors

This paper describes the effectiveness of an educational program aimed at health care professionals with the purpose to improve oral health in elderly at a long-term care institution. The topic is important since oral health is essential for the general health and quality of life of the elderly.  However, there is some weaknesses and unclearness with the current study that I will try to highlight below.

Title

The title is long but describe the study well.

Introduction

The introduction could be condensed and better disposed, see examples below.

Page 2, lines 49-50: The sentence “Healthcare providers have been found…” can be excluded since a deepening with the same content follows in the next paragraph.

Page 2, lines 65-66: The sentence “However, many studies only compare…” is a repetition of the same text in lines 58-59. I suggest that one of the sentences will be excluded.

Page 2, lines 68-74: Which study is being described? I miss a reference.

Page 2, lines 75-: Unclear what is meant by “…findings of this original study”.

Page 2, lines 78-80: This text is already mentioned above.

Page 2, line 88: I suggest to more clearly state that the aim applies to this follow-up study (and not the carefully described original study by writing: “Therefore, this follow up study aimed to…”.

The aim is hard to follow. Furthermore, aim is expressed differently in abstract and introduction. Is aim 1) describing the aim for the original study? And 2) aim for the follow up study? In line 89 the term “long-term impact” is used but in line 93 the term “short-term” is used. Please, explain this better.

Material and Method

This section is short and briefly describes the used method. The reader has to read reference 28 to understand how the study has been carried out. I have carefully read the Method section in reference 28 but still have some questions.

The sample: two LTCIs was chosen. How was the selection made? How was it decided which LTCI that should perform the intervention? Any randomized process or what? In the next step 20 professionals in each LTCI were chosen. How was this selection made? A randomized selection process can avoid bias such as differences in education level and experiences. Generalizability requires a representative sample.

A power calculation was made but no information of which expected effect that was used for the calculation or which study the power calculation was based on. This information could not be found in reference 28. No dropout seems to occur during the study. How did you manage to reach all 40 on four different occasions?! My experience is that the turnover among nursing staff in care for the elderly is high. 

The control group received “standard support”. What does that mean? I found no explanation in reference 28.

In reference 28 a second aim was “to assess the quality of oral care provided to residents by a small sample of healthcare workers after the OHE programme”.  What value has such a strongly limited sample without a control group in the evaluation of the OHEP?

I miss a description of the used statistical methods Mauchly´s and Tests of within-subject effects.

Results

Page 3, line128: I think the word “participating” should be used instead of “participated”.

Page 3, lines 130-132, table 1: Priority of daily duties are presented. In Method section this data collection is not described. I can´t find the information I references 28 and 32. Where did this knowledge come from. Was this collected on all four occasions or just at baseline? If data were collected at baseline the figures show important differences between the groups already before the intervention. Together with the baseline differences concerning knowledge, education level and experiences of taking care of the elderly, there were important differences between the groups from the start. Are the priority differences statistically significant?

Table 1 contains important data but is difficult to read. Is there any other design of table or figure that more clearly gives the reader a picture of the differences between the groups?

Table 2 seems to include two different analyzes? The authors may consider splitting the table into two. In the head of the table the words “pre- and postintervention” could be substituted with  “at different time points”. Since the table shows data from the control group it is not relevant to use post-intervention.

Table 3, head: “Intervention effects on knowledge….”The author can´t know that it is the intervention which has led to the change of data. The largest change in data over the period has occurred in the control group (for the worse) and this can´t be explained by the intervention. Alternative table head: “Knowledge, attitudes, and practice of healthcare providers in intervention and control group in 4 stages”.

The authors carry out a number of statistical analyzes to highlight differences between the intervention and control group. In summary, differences are found between the groups in terms of knowledge but not in terms of attitudes and practice. The improvement in knowledge in the intervention group was modest after the intervention and declined at 3- and 6-month follow-up. Therefore, the clear significant differences between the groups can probably be explained by the fact that the control group had better knowledge than the intervention group at the first three measurements. The control group showed a considerable deterioration in knowledge over time.

Discussion

I am not sure of how the authors use the term OHEP. Does the term refer to oral health educational programmes in general or is it this particular programme that was used in this study? In page 8, lines 251-252 I get the idea that the term is used for all educational programmes. It would be clearer to the reader if OHEP was used solely for this study´s program. 

It is good that the authors discuss possible reasons for the significant deterioration in the knowledge level of the control group but I do not find the factors highlighted convincing. Suspicion can be raised that the result is biased in some way, which may have to do with the differences between the groups at baseline in terms of education, experiences and priorities.  

Common to many studies is that the level of knowledge is relatively easy to raise, but it is considerably more difficult to transfer this to improved oral health for the elderly, see the systematic mapping of systematic reviews where the results from 11 systematic reviews describes the domain organisation of dental care for elderly (Ástvaldsdóttir Á, Boström AM, Davidson T, Gabre P, Gahnberg L, Sandborgh Englund G, Skott P, Ståhlnacke K, Tranaeus S, Wilhelmsson H, Wårdh I, Östlund P, Nilsson M. Oral health and dental care of older persons-A systematic map of systematic reviews. Gerodontology. 2018 Dec;35(4):290-304. doi: 10.1111/ger.12368. Epub 2018 Aug 20. PMID: 30129220).

It would be interesting if the authors could discuss the cost effectiveness of the intervention. The intervention is solid but quite extensive, thus it is important to reflect on the relation between benefits and use of resources.  

I agree that the study has some strengths as mentioned by the authors. I also think that the validated tool used for evaluation (ref 32) is a strength. The limited sample size entails difficulties in generalizing, but also the differences in the groups from the start contribute to this. Some form of randomization in the selection of institutions and participants could have improved this.

Conclusion

The conclusion about the effect of OHEP on the level of knowledge must be drawn with great caution given the unfavourable knowledge situation of the control group.

References

Journal titles should be abbreviated. Now there is a mix of abbreviations and full names.

The writing of year, volume and page range are not written equivalently.

Several websites are used as references. Dates for assessment are usually one year old. The links should be checked so they still work and a more recent date of assessment can be given.  

Author Response

(The authors gave the same response as above.)

Round 2

Reviewer 2 Report

Comments and Suggestions for Authors

This revised has been improved and most of mine comments has been taken care of.   

The shortened title is an improvement.

The introduction has been revised and now have a more logical flow. However, the introduction is still quite long.

The authors´ answers of my comments in the method section are satisfactory as is the revised text.

Lines 197-204: I don´t understand why certain text is highlighted green and others yellow. And why some words are italicized. 

Table 2 is improved.

Head of table 3: I still claim that the text “Intervention effects on knowledge…” should be changed to “Knowledge, attitudes and practice of healthcare providers in intervention and control group in 4 stages”. The authors have not shown that the intervention led to the change of data, only that the data have changed.

Author Response

Dear reviewer,

Thank you for your valuable comments again. Please see our replies below.

This revised has been improved and most of mine comments has been taken care of. 

Reply:  Thank you.

The shortened title is an improvement.

Reply:  Thank you.

The introduction has been revised and now have a more logical flow. However, the introduction is still quite long.

Reply:   Thank you. The introduction has been further shortened while maintaining the logical flow. We hope that this revised version will be acceptable.

The authors´ answers of my comments in the method section are satisfactory as is the revised text.

Reply:  Thank you.

Lines 197-204: I don´t understand why certain text is highlighted green and others yellow. And why some words are italicized. 

Reply: The different colors represent the revisions made by the authors at different periods. The specific part that was mentioned has been revised.

Table 2 is improved.

Reply:  Thank you.

Head of table 3: I still claim that the text “Intervention effects on knowledge…” should be changed to “Knowledge, attitudes and practice of healthcare providers in intervention and control group in 4 stages”. The authors have not shown that the intervention led to the change of data, only that the data have changed.

Reply:  Apology for missing this point. The heading of Table 3 and the description of data changes led by the intervention has been revised.